# Primary Amoebic Meningoencephalitis by *Naegleria fowleri*: Pathogenesis and Treatments

**DOI:** 10.3390/biom11091320

**Published:** 2021-09-06

**Authors:** Andrea Güémez, Elisa García

**Affiliations:** Facultad de Ciencias de la Salud, Universidad Anáhuac México Campus Norte, Huixquilucan 52786, Estado de México, Mexico; andrea.guemezga@anahuac.mx

**Keywords:** *Naegleria fowleri*, amoeba, pathogenesis, neurodegeneration, neuroinflammation, neuropathology, treatment

## Abstract

*Naegleria fowleri* is a free-living amoeba (FLA) that is commonly known as the “brain-eating amoeba.” This parasite can invade the central nervous system (CNS), causing an acute and fulminating infection known as primary amoebic meningoencephalitis (PAM). Even though PAM is characterized by low morbidity, it has shown a mortality rate of 98%, usually causing death in less than two weeks after the initial exposure. This review summarizes the most recent information about *N. fowleri*, its pathogenic molecular mechanisms, and the neuropathological processes implicated. Additionally, this review includes the main therapeutic strategies described in case reports and preclinical studies, including the possible use of immunomodulatory agents to decrease neurological damage.

## 1. Introduction

Several pathogens, including different bacteria, viruses, fungi, and parasites, have shown the ability to infect the human central nervous system (CNS) [1]. They have various molecular mechanisms that allow them to disseminate through the blood–brain barrier (BBB) or the blood-cerebrospinal fluid barrier (BCSFB) and take the brain and spinal cord as their primary target of infection [2].

Of those parasites capable of neuroinfecting, a group of opportunistic protozoa known as free-living amoebas (FLA) cause severe health problems. FLA are mitochondriate and eukaryotic microorganisms that can complete their life cycle as parasites or inhabiting natural environments as free-living amoeba. For this reason, FLA are known as amphizoic organisms [3,4]. These amoebas are ubiquitous and have been found in the air, soil, and water. However, they have also been identified in everyday objects like flower pots, humidifiers, sewers, swimming pools, water pipes, water parks, and even hospital environments [3].

*Naegleria fowleri* is one of the few FLA capable of causing fatal infections in humans. This parasite causes primary amoebic meningoencephalitis (PAM), an acute and fulminating infection that can lead to death 7 to 10 days after the amoeba enters the body. PAM is commonly observed in immunocompetent children and young adults, especially after having contact with amoeba-contaminated water. Even though the disease is extremely rare, it has a mortality rate of 98% [5,6]. Studies have suggested that an early diagnosis is critical for a patient to survive PAM [3]. However, due to its low morbidity, there is a lack of awareness and knowledge of the pathogenesis of the infection. This last issue makes it difficult to develop successful treatment and effective diagnostics tools [3]. Additionally, due to its non-specific symptoms, PAM is commonly unreported, unrecognized, or mistaken for bacterial or viral infections [4].

## 2. *Naegleria fowleri*

*Naegleria* is a genus of FLA that belongs to the family Vahlkampfiidae, order Schizopyrenida, and class Heterolobosea [7]. The genus consists of 47 species, which can be identified by variations in their genome, specifically their internal transcribed spacers (ITS) and their 5.8S rDNA. Out of all the *Naegleria* species, only *N. australiensis, N. italica*, and *N. fowleri* have been described as pathogenic. *N. australiensis* and *N. italica* only affect laboratory animals. However, *N. fowleri* is the only pathogen known to cause the fatal human disease primary amoebic meningoencephalitis (PAM) [8,9]. *N. fowleri* is most closely related to *N. lovaniensis*, although the latter is considered to be non-pathogenic [8].

*N. fowleri* is a thermophilic and ubiquitous amoeba that can be found in the air, soil, and warm waters [10]. Its natural habitats include hot springs, ponds, rivers, and freshwater lakes. However, it has also been identified in drinking water distribution systems, untreated swimming pools, fountains, hospitals, thermal waters, untreated drinking water, and water parks [11]. *N. fowleri* is a widely distributed parasite, since it has been identified in almost every continent, except Antarctica [10]. 

*N. fowleri* occurs in three different forms. When conditions become too hostile, the amoeba transforms into a metabolically inactive cyst, described as a spherical structure that measures from 7 to 12 µm in diameter, with a thick endocyst, a thin ectocyst, and some mucoid-plugged pores [12,13]. The cyst is incredibly resistant and can survive a variety of physical and chemical conditions, including temperatures as low as 4 °C; therefore, this amoeba can remain dormant during the cold winter months and reproduce during the summer [10].

When the amoeba faces non-nutritive conditions but is in the presence of water, it transforms into a transitory flagellate. This form has a pear-shaped appearance measuring from 10 to 16 µm and has two flagella of approximately the same length. They have a nucleus, a nucleolus, vacuoles, cytoplasmic inclusions, mitochondria, and a rough endoplasmic reticulum [12]. *N. fowleri* flagellate thrive at 27–37 °C, so they are usually present in warm waters or during the summer months [11,14].

Under favorable conditions, the amoeba can be found as a reproductively active trophozoite described as a long and slender structure that measures approximately 22 µm long and 7 µm wide. These trophozoites have a large nucleus with a nucleolus, many mitochondria, food vacuoles, a single contractile vacuole, an endoplasmic reticulum, ribosomes, and membrane-bound cytoplasmic organelles [10,12]. These also exhibit food-cup structures (amoebastomes) that have been related to their feeding [15]. The trophozoites are the only form of *N. fowleri* that can reproduce, feed, encyst, and cause infection in other organisms. Their primary food source comes from Gram-positive and Gram-negative bacteria, but trophozoites can also consume algae and yeast. They divide by binary fission and, being thermophilic, grow better at 35–46 °C [11].

## 3. Epidemiology and Clinical Manifestations of PAM

PAM is a hemorrhagic-necrotizing meningoencephalitis caused by *N. fowleri* and is seen mainly in immunocompetent children and young adults [12]. This infection occurs most frequently during the summer months when the water temperature is adapted to the thermal needs of the amoeba and people engage in recreational water activities [16].

Fowler and Carter were the first to describe PAM in 1965 after four people died in Australia’s Adelaide Children’s Hospital. The cause of death was attributed to an amoeba invading their meninges, which unleashed severe damage and inflammation in the brain [14,17,18]. Thenceforth, PAM has been reported in multiple countries, with an estimate of 400 victims worldwide. However, the total number of cases is unknown and could be greater due to misdiagnosis or unreported cases [3,4,16].

According to the most recent data, a total of 39 countries have reported cases of *N. fowleri* infections. However, the United States of America (USA), Pakistan, Mexico, Australia, the Czech Republic, and India have been the most affected. These countries may be more prone to these infections because of their year-round warm climates and access to contaminated water sources. It is worth mentioning that most USA cases happen in southern regions of the country, where the weather is warmer. The rest of the countries with documented cases of *N. fowleri* infections are shown in Figure 1 [11,19].

PAM is characterized by a mortality rate of 98% and, to date, no more than a dozen people have survived the infection [20,21]. This high mortality has been attributed to delays in diagnosis, a lack of safe and effective treatments, and the difficulty of delivering drugs to the brain [22]. Despite the fact that PAM presents a low morbidity rate, it has been postulated that climate change and increased temperatures could result in a higher frequency of *N. fowleri* infections [10,17]. 

PAM’s clinical manifestations usually appear from 5-to-7 days after the initial exposure but may develop after only 24 h [3]. These are usually indistinguishable from viral or bacterial meningitis, as patients present headaches, fever, nausea, fatigue, and vomiting [17]. During later stages, patients may have other signs and symptoms such as anorexia, irritation, nuchal rigidity, Kernig’s sign, Brudzinski’s signs, lethargy, photophobia, confusion, seizures, and possible coma [14,23]. People infected with *N. fowleri* usually die 1–2 weeks after the initial exposure, and because PAM has no specific clinical symptoms, patients are typically diagnosed post-mortem [14,24]. Autopsies have revealed that the cerebral hemispheres are soft, swollen, edematous, and gravely congested after an infection. The white matter of the brain and spinal cord exhibit focal demyelination. The olfactory bulbs present inflammatory exudates and hemorrhages, while the leptomeninges (arachnoid and pia mater) are congested, diffusely hyperemic, and with limited infiltration. Trophozoites, but not cysts, have been identified at the base of the brain, hypothalamus, midbrain, subarachnoid, and perivascular spaces [12,25]. Given the severity of the infection, early diagnosis and treatment are key factors for the patients’ survival. For an accurate diagnosis, it is essential to consider both CNS symptoms and a history of contact with contaminated water [12,14].

## 4. Diagnosis

During the initial stages of the infection, contrast-enhanced computerized tomography (CT) scan and magnetic resonance (MR) usually reveal cerebral edema, cortical sulcal effacement, and cisternal obliteration around the midbrain and the subarachnoid space. Once the infection progresses, these conditions typically worsen, revealing necrotic areas, stenosis, and aneurysms [26].

PAM can be officially diagnosed by obtaining cerebrospinal fluid (CSF) through a lumbar puncture, which will reveal many polymorphonuclear leukocytes and *N. fowleri* trophozoites [12,25]. CSF stained with Giemsa-Wright or trichrome, but not Gram, will reveal the presence of the amoeba and allow a morphological analysis of the parasite.

*N. fowleri* can be cultivated by transferring a few drops of CSF into a non-nutrient or low-nutrient agar plate seeded with living or heat-killed bacteria. However, it is preferable to use live bacteria such as *Enterobacter aerogenes*, *E. coli*, or other Gram-negative bacilli. The most recommended medium for *N. fowleri* is Nelson’s growth medium, which comprises Page’s amoeba saline (0.4 mg of MgSO_4_, 0.4 mg CaCl_2_, 14.2 mg Na_2_HPO_4_, 13.6 mg KH_2_PO_4_, 12 mg NaCl in 100 mL of distilled water), along with 0.17 g liver infusion and 0.17 g glucose. Before usage, the medium should always be autoclaved for 25 min at 121 °C, followed by incorporating sterile, heat-inactivated, fetal calf serum [12]. Additionally, supplementing Nelson’s growth medium with 1% peptone has shown to improve the amoeba’s growth [9].

The culture should be incubated at 37 °C and monitored every day during a week. Trophozoites generally develop within the first three days and, as their food source declines, begin to encyst after 7 to 10 days [9]. To transform the trophozoites into flagellates, one must combine one drop of the amoebae culture or sedimented CSF with one mL of distilled water during 1-to-2 h. This flagellation process can help distinguish *N. fowleri* from other pathogenic amoebae [12].

Even though solid non-nutrient agar is the standard method for growing *N. fowleri*, it has limitations, such as bacterial contamination and a lower yield of cysts. Recent studies have suggested a liquid encystment medium, modified from Page’s amoeba saline, as a replacement. The proposed method involves cultivating the amoeba in Nelson’s growth medium supplemented with 10% fetal bovine serum and incubating it at 37 °C. Once the trophozoites develop, about 2 × 10^6^ cells are washed twice in PBS (pH 7.4) and incubated in 24-well plates with 5 mL of the encystment medium at 37 °C. The medium is comprised of 120 mM NaCl, 0.03 mM MgCl_2_, 1 mM NaHPO_4_, 1 mM KH_2_PO_4_, 0.03 mM CaCl_2_, 0.02 mM FeCl_2_, pH 6.8. The previous method managed to encyst the trophozoites after 48 h and could be an alternative way to obtain *N. fowleri* cysts [27].

Further confirmation of the amoeba’s presence may be done through microscopy, immunofluorescence assay (IF), enzyme-linked immunosorbent assay (ELISA), or flow cytometry (FC) [12]. Furthermore, a reverse transcription polymerase chain reaction (RT-PCR) is recommended to determine the amoeba’s genus and species [26].

Additionally, intracranial and CSF pressure usually increases to 600 mm H_2_O or higher, a symptom that has been directly associated with the patient’s death. CSF’s color may vary from greyish to yellowish-white during the early stages of the infection, but as the disease progresses, it may turn red as a consequence of the large number of red blood cells (RBCs), which have been reported as high as 24,600/mm^3^. Furthermore, white blood cells (WBCs) count ranges from 300 cells/mm^3^ to 26,000 mm^3^, protein concentration may vary from 100 mg/100 mL to 1000 mg/100 mL, while glucose may present values equal or lower to 10 mg/100 mL [13,23].

Clinical features, combined with the previously reported symptoms and a recent history of contact with water, confirm infection with *N. fowleri*. At that point, it is crucial that the patient receives medical treatment rapidly to improve the probability of survival [12,14].

## 5. Pathogenesis of PAM

More than 90% of infections by *N. fowleri* occur when people submerge, dive, or splash in warm water bodies, allowing the amoeba’s trophozoites to enter the body through the nasal cavity [23,28]. Even though the infection usually originates through the practice of recreational aquatic activities, it can also occur through ablution practices performed by religious groups and hygiene devices like neti-pots [12]. Additionally, it has been suggested that the amoeba can “dry-infect” through cyst-laden dust, causing infection after transforming into their trophozoite form. Although the latter mechanism accounts for only about 6.5% of PAM cases, it is extremely concerning, as little can be done to prevent the inhalation of dust [10]. Once the amoeba is inside the nasal cavity, it attaches itself to the nasal mucosa, penetrates it, and migrates along the olfactory nerves through the cribriform plate until it reaches the olfactory bulb. It then enters the brain through olfactory nerve bundles, where it multiplies and causes cerebral edema, herniation, and eventually death (Figure 2) [12,23,28,29].

*N. fowleri* causes severe damage to the CNS because of the amoeba’s pathogenicity and the intense immune response it unleashes [12,23]. There is limited information on *N. fowleri’s* virulence factors, but in vivo, in vitro, and ex vitro models have been developed to understand the molecular mechanisms associated with the pathogenesis of PAM [12,25]. Thus far, two pathogenic mechanisms have been recognized in *N. fowleri:* Contact-dependent mechanisms related to adhesion and phagocytic food-cups and contact-independent mechanisms involving cytolytic molecules secreted by the amoeba [31].

### 5.1. Contact-Dependent Mechanisms

One of the most critical steps in any microbial infection is the adherence of the pathogen to its host cell [32]. It has been suggested that pathogenic amoeba like *N. fowleri* present a higher attachment level than nonpathogenic species. In vitro studies have shown that *N. fowleri* can move through the nasal epithelium adhering to basement membrane components, such as laminin-1, collagen I, and fibronectin [33]. This adhesion process is considered to be mediated by integrin-like proteins co-localized with actin filaments (53, 70 kDa) and fibronectin-binding proteins (60 kDa) that have been identified in the amoeba [32,33]. Additionally, kinases C found in *N. fowleri* have shown to increase the amoebic adhesion and cytotoxicity towards the host cell [32,33].

It has been suggested that the adhesion of *N. fowleri* to the components of the extracellular matrix (ECM) could activate signaling transduction pathways that trigger the expression of specific proteins and proteases that facilitate the entrance and proliferation of amoeba in the CNS. An in vivo experiment demonstrated that axenically maintained *N. fowleri* amoebae are mildly pathogenic. However, when passed through the brain of a mouse, they became almost 100-fold more virulent, and exhibit a group of proteins linked with cytoskeletal reorganization and stability. A particular protein homologous to Rho guanine nucleotide exchange factor 28 was found in highly virulent *N. fowleri,* suggesting that proteins associated with cytoskeletal rearrangement and stabilization are involved in the amoeba’s invasion and pathology [15].

In addition to adhesion, amoebas cause severe tissue destruction through contact-dependent phagocytosis, where they use their amoebastomes to gradually engulf neuronal cells [12,23,34]. It has been suggested that weakly pathogenic strains destroy nerve cells by ingesting them with their food-cup structures. In contrast, highly pathogenic amoebas lyse cells on contact and feed on the resulting cell debris [35]. The Nfa1 gene found in *N. fowleri*, which encodes for the Nfa1 protein (13.1 kDa), has been linked to the amoeba’s locomotion and amoebastome formation. Studies have shown that the use of anti-Nfa1 antibodies decreases the cytotoxic effects of the amoeba, suggesting that this protein plays an essential role in its contact-dependent pathogenesis [34]. Additionally, transfection of *N. fowleri’s* Nfa1 gene into the non-pathogenic *N. gruberi* enhanced the cytotoxicity of the parasite towards Chinese hamster ovary cells (CHO), compared to naïve *N. gruberi* [36]. Furthermore, an Nf-actin protein (50.1 kDa) encoded by the Nf-actin gene has been identified in the amoeba’s cytoplasm, pseudopodia, and amoebastomes. This protein has been related to the increase of *N. fowleri’s* cell adhesion, phagocytosis, and cytotoxicity [31]. Furthermore, another study has revealed a membrane protein called Mp2CL5 (17 kDa) in pathogenic *Naegleria* species. It has been proposed that it plays a role in the amoeba’s cellular recognition and adhesion [37].

### 5.2. Contact-Independent Mechanisms

Matrix metalloproteinases (MMP) are endopeptidases that have been linked with the invasion of different parasites and leukocytes into the CNS. Recently, a study has confirmed that *N. fowleri* trophozoites have at least three of these endopeptidases, specifically MMP-2, MMP-9, and MMP-14. MMP-2 and MMP-9 are specialized in cleaving gelatin and type IV collagen. However, being proenzymes, they must be previously activated by MMP-14 to function. The results of this study suggest that *N. fowleri* secretes these MMPs and degrades the ECM, simplifying its passage from the nasal chambers into the olfactory bulb [38].

It has been suggested that *N. fowleri* can cross through the BBB by degrading its tight-junction proteins (TJP). An in vitro study has shown that the amoeba secretes cysteine proteases that delocalize and degrade TJP, such as claudins-1 and occludins (ZO-1), as well as modifying the actin cytoskeleton [39]. This process alters the stability of the BBB and allows the amoeba to cross into the CNS successfully [2]. In addition, it has been hypothesized that *N. fowleri* also uses cysteine proteins to degrade iron-binding proteins, allowing the amoeba to obtain iron while inside the host. However, the authors of this study suggest that more experiments should be performed to further confirm this statement [40]. Furthermore, a 30 kDa cysteine protease secreted by *N. fowleri* exhibited cytopathic effects towards the kidney cells of baby hamsters (BHK), while a 37 kDa cysteine protease that presented mucinolytic activity is thought to be involved in mucin degradation and evasion of the host’s immune system [41]. Other cysteine proteases with molecular weights of 58 kDa, 128 kDa, and 170 kDa have also been found in *N. fowleri* [12]. In addition, cathepsin B (NfCPB) and cathepsin B-like (NfCPB-L) cysteine proteases with a molecular weight of 38.4 kDa and 34 kDa, respectively, are supposed to participate in proteolytic activities on immunoglobulins, collagen, fibronectin, hemoglobin, and albumin [42].

Other molecules related to causing cell lysis and cytotoxicity in *N. fowleri* have been identified, such as the pore-forming protein abbreviated as N-PFP (66 kDa). This is a membrane-bound protein that can lyse nucleated cells and affect the integrity of the host cell membrane by depolarizing it [43]. Additionally, two pore-forming peptides belonging to the protein family of saposin-like proteins (SAPLIP), called *Naegleriapores* A and B, have been identified in highly virulent *N. fowleri*. Both molecules are isoforms and have structural properties that resemble those of antimicrobial and cytolytic polypeptides found in other virulent amoebas, as well as human Natural Killer cells (NK) and T cells. These exhibit powerful pore-forming activity and can kill both prokaryotic and eukaryotic cells [12,44].

Different studies have reported the activity of several phospholipases (A, A^2^, and C), sphingomyelinases, neuraminidases, elastases, and proteolytic enzymes, which are believed to be used by the amoeba to lyse host cells. Moreover, phospholipases, lysophospholipases, and sphingomyelinases have also been related to the damage found in the cytoplasmic membranes of cells and the demyelination observed in the white matter of patients with PAM. This suggests that the molecules could be important factors in the neurodegenerative process. On the other hand, *N*-acetylglucosaminidase, acid phosphatase, 5′-nucleotidase, aspartate aminotransferase, alpha-d-glucosidase, beta-glucosidase, beta-galactosidase, beta-fucosidase, alpha-mannosidase, hexosaminidase, arylsulfatase A, beta-glucuronidase, electron-dense granules, peroxiredoxin, and thrombin receptors have been identified in the amoeba and may play a role in its pathogenesis [12]. Nevertheless, its pathogenic mechanisms are not entirely understood [45].

*N. fowleri* is capable of producing nitric oxide (NO) in vitro through an isoform of a nitric oxide synthase (NOS) that shares epitopes with the mammalian NOS; despite this, the enzyme remains unidentified. The same study also concluded that *N. fowleri* presented higher resistance to NO toxicity, which could explain their ability to be unaffected by the inflammatory response of the host induced in the olfactory bulb [46,47]. Increased levels of a heat shock protein 70 (HSP70) weighing 72 kDa has also been identified in highly virulent *N. fowleri* trophozoites [15,48]. HSP70 has been related to the amoeba’s ability to resist cellular stress and temperature changes. Besides, an in vitro study suggests this protein also plays a role in the proliferation, cytotoxicity, and regulation of the pathogen in the host immune system [48].

Moreover, a study using genomic and proteomic approaches to identify variations in the protein expression of highly and weakly virulent *N. fowleri* trophozoites, recognized other molecules that could be related to the amoeba’s pathogenic mechanisms. The apoptosis-linked gene-2-interacting protein X1 (AIP1) is thought to be a crucial regulator of endosomal sorting, a system that executes intracellular transportation of cellular material between organelles. Furthermore, the Golgi transmembrane protein HID-1 has been associated with vesicular exocytosis. Both proteins were over-expressed in highly pathogenic trophozoites, regulating vesicular transportation. The Ras-related protein Rab-1 was also up-regulated in highly virulent amoebas and was linked to vesicular trafficking and the process of phagocytosis. Other molecules such as myosin II heavy chain, myosin Ie and villin-1 protein were identified in the amoeba and are thought to influence phagocytosis of target cells. Finally, a cyclophilin was overexpressed in highly pathogenic trophozoites as a pathogenic molecule [12,49]. 

## 6. Neuroinflammation and its Association to PAM Pathogenesis

*N. fowleri* may promote further host injury through robust induced immune response [50]. In vivo studies have demonstrated that trophozoites reach the olfactory bulb just 72 h post-infection (PI), although the first sign of tissue destruction and inflammation are seen after 96 h PI [50,51]. Studies have shown that *N. fowleri* trophozoites induce the production of reactive oxygen species (ROS), which in turn activate the epidermal growth factor receptor (EGFR) pathway and induce the expression of MUC5AC (mucin), and the pro-inflammatory cytokine, interleukin 8 (IL-8), which is one of the most powerful neutrophil chemoattractants. ROS also induce the expression of the pro-inflammatory cytokine IL-1β through an EGFR-independent mechanism [52]. An in vitro study suggested that ROS stimulate the formation of the NLRP3 inflammasome, a multiprotein complex capable of activating caspase-1 and secreting an active IL-1β [53]. In addition, an in vitro study demonstrated that *N. fowleri* activates ROS-dependent programmed necrotic cell death (necroptosis) in Jurkat cells. The authors suggest that necroptosis may be a defense mechanism against the amoeba; however, representing a high cost, cell destruction [54].

Following 102 h PI, a few eosinophils and neutrophils begin to surround the trophozoites in the olfactory bulb. Eosinophils can produce various pro-inflammatory cytokines and chemokines such as tumor necrosis factor alpha (TNF-α), IL-6, IL-8, and eotaxin. Nevertheless, eosinophils seem to be unable to remove the parasite in vivo, which may increase leukocyte recruitment and inflammation during the later stages of the infection. During 108–120 h PI, the number of eosinophils seems to decrease while neutrophils and macrophages increase [50]. Neutrophils can eliminate various pathogens through different mechanisms, including degranulation, proteolytic enzymes, antimicrobial peptides, ROS, and reactive nitrogen species (RNS). An in vitro study recently demonstrated that *N. fowleri* induces the production of neutrophil extracellular traps (NETs), which consist of nuclear or mitochondrial DNA combined with histones and protein components of cytoplasmic granules. However, this study also showed that the amoeba was efficiently killed by neutrophils when it was opsonized by immunoglobulin A and G (IgA and IgG) but evaded killing when it was unopsonized [55]. Additionally, it has been suggested that neutrophils require the presence of TNF-α to destroy *N. fowleri*, although evidence has demonstrated that this pro-inflammatory cytokine has no effect on the amoeba [35]. Extensive tissue damage distinguished by lytic necrotic areas, hemorrhages, and cellular debris has been detected in infected tissues. However, the damage has been associated with the presence of inflammatory cells [50].

It has been suggested that *N. fowleri* releases cysteine proteases to cross through the BBB; however, the damage caused to this structure also facilitates the passage of immune cells into the brain. The trophozoites have demonstrated the ability to induce extensive production of leukocyte adhesion molecules, such as vascular cell adhesion molecule 1 (VCAM-1) and intracellular adhesion molecule 1 (ICAM-1). These, as well as the disruption of the BBB, have been associated with a high number of inflammatory cells entering the CNS. Besides its NO production, *N. fowleri* can induce the production of this compound by the cerebrovascular endothelium, perhaps through an interaction between cytokine receptors with the endothelium’s toll-like receptors (TLR), specifically TLR4, and activating the endothelial NOS (eNOS) as well as the inducible NOS (iNOS). The NO released may further alter the permeability of the BBB, allowing more leukocytes to enter the CNS [39]. Macrophages have also been shown to release TNF-α, IL-1, and NO as a mechanism to destroy different pathogens, although research has demonstrated that *N. fowleri* presents a high tolerance for NO toxicity [35,47]. Even though NO is a potent antimicrobial agent, it also has the potential to damage the host since its reduced form can produce peroxynitrite, a strong neurotoxin [56]. Actually, it has not yet been determined whether NO truly contributes to tissue damage in *N. fowleri* infections [47].

The brain tissue is composed of different types of cells, and microglia is one of them. These cells are the first line of defense within the CNS and can perform antigen-presenting and pro-inflammatory functions when activated by injury or infection [57]. In vitro, *N. fowleri* trophozoites activate microglial cells inducing the production of pro-inflammatory cytokines such as IL-1β, IL-6, TNF-α, ROS, and RNS [50,58]. It has been suggested that this activation occurs through the interaction of the microglia’s TLRs with the amoeba’s pathogen-associated molecular patterns (PAMPs) as well as the danger-associated molecular patterns (DAMPs) released by tissue damage [58,59]. In fact, microglial activation may cause severe damage to the CNS when it releases high doses of cytokines, ROS, and RNS, as these molecules can be highly neurotoxic [58]. Furthermore, high ROS levels have also been related to causing cell destruction through lipid peroxidation [60].

The amoeba also promotes the activation of astrocytes, CNS cells involved in maintaining the homeostatic environment and regulating the immune system within the brain. *N. fowleri* lysates can induce the expression of IL-1β and IL-6 in primary rat astrocytes through the activation of the activator protein 1 (AP-1) transcription factor. The expression of these cytokines was also dependent on the activation of the extracellular-signal-regulated kinase (ERK), the c-Jun N-terminal kinase (JNK), and p38 mitogen-activated protein kinase (MAPKs) pathways [61]. It has been proposed that the production of these pro-inflammatory cytokines in response to *N. fowleri* are not beneficial and may contribute to brain tissue destruction through an immunopathological process. Indeed, the release of pro-inflammatory cytokines stimulates the further breakdown of the BBB and induces the hyper inflammation of the brain with immune cells from non-neural sites [35]. Most of the tissue damage in human autopsies can be seen in the frontal areas of the brain, where inflammation is more intense. On the other hand, the posterior parts of the brain show no inflammation or tissue destruction despite revealing the presence of trophozoites. This suggests that inflammation plays a vital role in brain tissue damage in patients suffering from PAM [50].

## 7. Potential Treatment

PAM is a rare, acute, and fulminating infection characterized by very low morbidity and a very high mortality rate, two factors that complicate the development of a treatment [25]. Currently, all the information regarding potential drugs is obtained from a few case reports and different in vitro and in vivo studies; however, because of the infection’s infrequency and rapid progression, there is scarce possibility to perform clinical trials to test their safety and efficacy [23].

Nowadays, the most used drug to treat *N. fowleri* infections is Amphotericin B (AmB), an antifungal agent that has proven to kill amoebas by inducing an apoptosis-like programmed cell death (PCD) [62]. In the past, the interveinal or intrathecal administration of AmB alone or in combination with other drugs like Fluconazole (FCZ), Azithromycin (AZM), and Rifampin (RIF) has shown some efficacy in eliminating the amoeba, when delivered during the early stages of the infection [23,26]. However, high concentrations of these drugs must be administered to reach the minimum inhibitory concentration (MIC) required to kill the amoeba inside the CNS, as they have exhibited poor penetration of the BBB [63]. Furthermore, AmB is water-insoluble and is known to cause renal toxicity, anemia, chills, nausea, fever, vomiting, and headaches, which is why a safer and more efficient alternative is needed [62,64]. On the other hand, Miltefosine (MLT), an alkylphosphocholine compound used to treat breast cancer and leishmania infections, was considered a novel therapeutic drug to treat PAM when it successfully treated a 12-year-old girl who survived the infection with a full neurological recovery. In addition to the administration of MLT, AmB, FCZ, AZM, RIF, and Dexamethasone (DEX), the young girl was induced into a hypothermic state, which proved to manage intracranial pressure and reduce brain injury caused by hyper inflammation [65,66]. This treatment regimen was later used on two different patients, where one of them survived with a poor neurological state and the other one passed away. While MLT is a promising drug to treat PAM, it does not guarantee a full recovery [67].

### 7.1. In Vitro Studies

Several in vitro studies have recently been determined to find safer and more effective drugs to treat PAM patients (Table 1). However, since *N. fowleri* infections have a low morbidity rate, there is little financial incentive for the pharmaceutical industry to develop an anti-PAM drug. Consequently, the most cost-effective method to discover a treatment against this amoeba is through drug “repurposing”. Just like pathogenic fungi, *N. fowleri* uses ergosterol as a building block for its membrane. Therefore, anti-mycotic drugs that disrupt sterol biosynthesis may be repurposed to treat the amoeba. A study proposed utilizing a group of anti-fungal drugs known as conazoles to inhibit the sterol 14-demethylase (CYP51) and block sterol biosynthesis in *N. fowleri*. All seven conazoles (Itraconazole, Posaconazole, Ketoconazole, Isavuconazole, Miconazole, Voriconazole, Clotrimazole, and Fluconazole) demonstrated anti-amoebic activity in vitro, even exceeding the potency of MLT. However, only Itraconazole and Posaconazole displayed a half-maximum effect concentration (EC_50_) ≤ 0.01µM, more potent than AmB. Posaconazole managed to inhibit *N. fowleri’s* growth by 40% after 16 h post-exposure and up to ~90% after 24 h. Even though Posaconazole demonstrated potency against the amoeba, the drug has shown low brain permeability, a disadvantage for treating PAM [68].

Additionally, the amoeba’s sterol pathway can be blocked by Isavuconazole, Epiminolanosterol, and Tamoxifen, which inhibit CYP51, sterol C24-methyltransferase (SMT), and sterol 87-isomerase (ERG2), respectively. Combinations of these three drugs have shown synergistic effects against *N. fowleri* in vitro, hindering its growth by 95% 48 h after treatment. Therefore, inhibiting two enzymes involved in the sterol pathway may be more effective than inhibiting one [69].

Another study reported the amoebicidal activity of Ebselen, BAY 11-7082, and BAY 11-7085, over *N. fowleri*. First, these compounds were chosen as potential agents against the deadly amoeba due to their ability to cross through the BBB and their capability to inhibit cysteine proteases, which have been described as a virulent factor in *N. fowleri.* This drug exhibits anti-inflammatory activity, which may help decrease tissue damage caused by immune cells. The in vitro study showed that Ebselen inhibited the growth of *N. fowleri* trophozoites by 100% at a concentration of 12.5 µM while BAY 11-7082 and BAY 11-7085 showed 100% growth inhibition at a concentration of 3.12 µM. Additionally, it was reported that the (EC_50_) of Ebselen, BAY 11-7082, BAY 11-7085, and MLT obtained by the ATP bioluminescence method were approximately 6.2, 1.6, 2.3, and 54.5 µM, respectively. The compounds also demonstrated more potency than MLT by 8.5-fold, 34-fold, and 23-fold, respectively, making them potential new drugs to treat PAM [70].

Auranofin (AF) is a Food and Drug Association (FDA)-approved drug to treat rheumatoid arthritis. A recent in vitro study described its capacity to inhibit the growth of five different *N. fowleri* strains, with more potency than MLT but less than AmB. The concentration of 12.5 µM of AF inhibited the growth of the trophozoites by 97% after 24 h, and the drug showed no cytotoxicity to human astrocytes at concentrations lower than 5 µM. AF is a potential new drug to treat PAM due to its anti-inflammatory properties, its capacity to penetrate the BBB, and synergistic activity with AmB [22].

Staurosporine (STS), an indolocarbazole isolated from the bacteria *Streptomyces sanyensis,* has also demonstrated high activity against *N. fowleri* trophozoites in vitro. It has been suggested that its amoebicidal activity occurs by inhibiting the amoeba’s protein kinase (PK) and inducing an apoptosis-like mechanism via the mitochondrial pathway, but, more studies are required to confirm the mechanism of action. The natural compound demonstrated a half-maximum inhibitory concentration (IC_50_) value of 0.08 µM for two different *N. fowleri* strains and showed lower levels of cell cytotoxicity against murine macrophage J774A.1 than MLT [71].

A different study demonstrated the efficacy of Etomoxir (ETO), Perhexiline (PHX), Valproic acid (VPA), and Thioridazine (TDZ) against *N. fowleri* trophozoites. Although ETO, PHX, and TDZ inhibited the growth of the amoeba, it was TDZ that exhibited the lowest IC_50_ value (6–10 µM). Moreover, the study revealed additivity when combining MLT with PHX, VPA, and TDZ individually, and synergy between ETO and MLT. It was suggested that ETO could be included in the current treatment regime to reduce MLT concentrations [72].

Several investigations have recently focused on the anti-amoebic activity of statins over *N. fowleri* trophozoites. One particular study examined the in vitro activity of Simvastatin, Fluvastatin, Atorvastatin, Pravastatin, Mevastatin, and Lovastatin over two strains of the amoeba. Out of the six molecules, Fluvastatin was the most active, with IC_50_ values of 0.179 µM and 1.682 µM for the ATCC^®^ 30808 and ATCC^®^ 30215 strains, respectively. Atorvastatin also demonstrated activity against the strains, with IC_50_ values of 7.62 µM and 6.27 µM. However, the rest of the statins exhibited higher IC_50_ values, ranging from 19 µM to 34 µM. Furthermore, Pravastatin and Mevastatin showed no amoebicidal activity [24].

Even though Fluvastatin and Atorvastatin exhibited lesser potency but similar cytotoxicity to AmB, they might present an advantage against the anti-fungal drug because of their ability to penetrate the BBB [24]. Further investigations demonstrated that these statins showed higher induction of PCD in *N. fowleri* and lower cellular toxicity than MLT, suggesting that these drugs could be potentially used as a novel therapy for PAM [73]. 

Pitavastatin is another statin that shows potential against *N. fowleri* trophozoites. This drug has demonstrated EC_50_ values ranging from 0.3 µM to 4 µM against five different strains, proving to be more potent than MLT (EC_50_ ranging from 15–58 µM) but less than AmB (EC*_50_* ranging from 0.06–0.2 µM). In vitro, Pitavastatin inhibits the amoeba’s development by more than 60% as soon as 10 h after exposure. After 16 h, inhibition reaches 81%, and at 24 h, it comes to 96%. In addition, Pitavastatin’s action against different human cell lines is extensively documented, as it is an FDA-approved cholesterol-lowering medication. In this study, Pitavastatin did not cause toxicity in human T cells when evaluated at 10 µM; however, it exhibited an EC_50_ of 20 μM to liver hepatocellular carcinoma cells (HepG2) and Human embryonic kidney 293 cells (HEK293). Finally, unlike AmB, Pitavastatin can cross through the BBB, making it a potential candidate for treating PAM [74].

A subsequent study evaluated the anti-amoebic activity of Cerivastatin, Rosuvastatin, and Pitavastatin against two strains of *N. fowleri*. Cerivastatin demonstrated the highest activity against the ATCC^®^30215 strain presenting an IC_50_ value of 0.064 µM. On the other hand, Pitavastatin was the most potent against the ATCC^®^ 30808 strain, with an IC_50_ value of 0.038 µM. However, Rosuvastatin showed to be the least cytotoxic towards Murine macrophage J774.A1 cells. Additionally, Cerivastatin and Pitavastatin demonstrated higher activity than Fluvastatin and Atorvastatin against *N. fowleri*, as well as BBB permeability, making these statins potential anti-amoebic drugs [75].

Lonafarnib is another drug that has been evaluated against *N. fowleri* trophozoites, both individually and combined with Pitavastatin. On its own, Lonafarnib exhibited variated potency against five different *N. fowleri* strains, with EC_50_ values ranging from 1.5 µM to 9.2 µM. However, its combination with Pitavastatin showed synergistic activity, resulting in a 95% growth inhibition of the trophozoites. Additionally, Lonafarnib has proved to be well tolerated in different clinical studies and is thought to be BBB permeable. Therefore, the combination of Lonafarnib and Pitavastatin could be a potential treatment against *N. fowleri* [76].

Although the previously described molecules have demonstrated anti-amoebic activity in vitro, it is necessary that in vivo studies take place to further understand their true potential as therapeutic drugs against PAM.

### 7.2. In Vivo Studies

Some other drugs have shown amoebicidal activity in vivo (Table 2). One study demonstrated the effect Chlorpromazine (CPZ) had over *N. fowleri* trophozoites, and compared it to AmB and MLT. In vitro, AmB and CPZ managed to inhibit the growth of the parasite by 100% at concentrations of 0.78 and 12.5 µg/mL, respectively, while MLT sustained growth inhibition for six days at 25 µg/mL in vitro. Although AmB showed the best amoebicidal activity in vitro, CPZ was the most effective drug in vivo, presenting a 75% survival rate in mice. In comparison, AmB and MLT showed survival rates of 40% and 55%, respectively. Additionally, no liver or kidney toxicity was observed in mice treated with CPZ. However, further studies are required to understand the mechanism of action of this drug [77].

A different study evaluated the effects of seven antibiotics on *N. fowleri*. It was reported that Hygromycin B (Hy) and Rokitamycin (RKM) were the most effective in vitro, inhibiting the growth by 100% at concentrations of 12.5 and 6.25 µg/mL, respectively. At the same time, Roxithromycin (ROX) maintained growth inhibition at 25 µg/mL. However, Clarithromycin, Erythromycin, Neomycin, and Zeocin did not exhibit any amoebicidal activity. Afterward, RKM exhibited a survival rate of 80% in infected mice, while ROX presented a survival rate of 25%. Curiously, even though Hy showed promising results in vitro, no infected mouse survived when treated with this drug. Although RKM’s mechanism of action is still unclear, it is a good candidate to treat PAM. It has shown amoebicidal effects against *N. fowleri* and presented no liver or kidney toxicity in the infected mice [78]. 

Corifungin, a water-soluble polyene macrolide, has demonstrated more potent activity against *N. fowleri* than AmB. A study reported that this drug inhibited the growth of the pathogenic amoeba by 100% at a concentration of 25 µM. Moreover, the administration of Corifungin at 9 mg/kg/day for ten days resulted in a survival rate of 100% of the infected mice and was well tolerated by the animals. On the other hand, the same dose of AmB showed a survival rate of 60% and was similar to the phosphate-buffered saline (PBS) controls. Corifungin may be able to penetrate the BBB more effectively than AmB perhaps due to its water solubility, but more studies must be carried out to prove it. Nonetheless, the FDA approved an orphan drug designation to Corifungin to treat PAM [64]. 

A more recent study performed a high-throughput phenotypic screening method and identified various antibiotics and antifungals that could be repurposed to treat PAM. The most outstanding drug was the antifungal Posaconazole (PCZ). This compound inhibits the growth of the amoeba in just 12 h and showed an IC_50_ value of 0.24 µM. PCZ also demonstrated additive activity with AZM, AmB, and MLT, although no synergy was observed. In addition, the intravenous administration of 20 mg/kg of PCZ showed a survival rate of 33% in infected mice, while the combination of PCZ with AZM significantly prolonged survival [79].

### 7.3. Nanoparticle Conjugation

One of the main problems surrounding CNS infections is the low efficacy many drugs exhibit due to their inability to penetrate the BBB successfully. In some cases, this problem is solved by increasing the dose of the drug administered, but this may lead to cell toxicity [26]. In more recent years, nanoparticles have gained attention in the pharmaceutical industry for their ability to enhance the efficacy of a drug, which allows a decrease in dosage [63]. Since nanoparticle drug delivery systems have been demonstrated to increase bioavailability, decrease cell toxicity, and are site-specific, there have been a few studies determined to understand the advantages of using drugs conjugated with nanoparticles to treat PAM (Table 3) [80].

It has recently been demonstrated that AmB and Nystatin (NYS) conjugated with silver nanoparticles (AgNp) are more effective against *N. fowleri* than the drugs alone. In this particular study, unconjugated AmB and NYS reduced the number of *N. fowleri* from 9.2 × 10^5^ to 3.7 × 10^5^ and 7.5 × 10^5^, respectively. However, AmB and NYS conjugated with AgNp reduced the number of amoebas to 2 × 10^5^ and 5.8 × 10^5^, respectively. Not only does AgNp increase the drug’s amoebicidal activity, it also reduced cell cytotoxicity caused by the trophozoites [63]. 

A different study synthesized 34 aryl quinazoline derivatives and tested their amoebicidal activity against *N. fowleri* with and without AgNp conjugation. Several aryl quinazolinones reduced the viability of the trophozoites on their own, and in some cases their activity was enhanced by AgNp conjugation. This was most noticeable with quinazolinone 24 (Q24), which presented an amoebicidal activity against *N. fowleri* of 37% alone and 53% while conjugated with AgNp [81]. On the other hand, oleic acid (OA) has also been conjugated with AgNp to study its activity against the amoeba. The study concluded that 10 µM of OA-AgNp was able to reduce the number of *N. fowleri* from 6.1 × 10^5^ to 2.1 × 10^5^. This conjugation exhibited higher activity against the trophozoites than AmB, which reduced the amoeba to 2.7 × 10^5^ [80]. Another study investigated the conjugation of three different FDA-approved drugs to reduce the number of *N. fowleri* trophozoites. While Diazepam (DZP), Phenobarbitone (PBT), and Phenytoin (PTN) are commonly used to treat anxiety and control seizures, the study hypothesized a secondary application as an antiparasitic drugs because they can penetrate the BBB. All three drugs exhibited amoebicidal activity against *N. fowleri,* and more potency when conjugated with AgNp, but only PTN proved to be as effective as AmB [82].

Furthermore, six novel azole derivatives (A1-A6) were synthesized and tested in vitro against *N. fowleri*, both independently and conjugated with AgNp. This study concluded that in concentrations of 50 µM, the azoles showed both amoebicidal and amoebistatic activity against the trophozoites. A4 and A5 azoles exhibited the highest amoebicidal activity, resulting in 72% and 66% cell death, respectively. On the other hand, azole A3 showed the highest amoebistatic activity, causing 75% growth inhibition. Interestingly, nanoparticle conjugation slightly improved the amoebicidal activity of the molecules, as it only increased azole A1 by 36% and azole A6 by 6% [83].

A similar study synthesized four novel bisindole and thiazole derivatives (A1–A4) and tested their in vitro activity against *N. fowleri*. A4 and A2 azoles showed the most amoebicidal activity, resulting in a percentage cell death of 69% and 53%, respectively. On the other hand, A4, A3, and A1 azoles resulted in 68%, 61%, and 49% amoebistatic activity. Afterward, the azoles were conjugated with AgNp to increase their anti-amoebic activity. However, the conjugation only boosted the activity of A1 and A3 by 32% and 51%, respectively [84].

Gold nanoparticles (AuNp) have also been studied as potential drug delivery systems to treat PAM. One study conjugated AuNp with curcumin to evaluate its capacity against *N. fowleri,* since the molecule presents anti-inflammatory properties and can inhibit lipid peroxidation. Curcumin demonstrated a concentration-dependent activity against *N. fowleri,* resulting in a 66% amoebicidal activity at a concentration of 200 µM. However, conjugation with AuNp enhanced curcumin’s bioavailability significantly, resulting in a 69% amoebicidal activity at a concentration of 10 µM. In addition, curcumin conjugated with AuNp did not exhibit cytotoxic activity against human keratinized skin cells (HaCaT) [85]. Furthermore, AuNp has also been conjugated with trans-cinnamic acid (CA), a secondary metabolite obtained from plants. Since CA exhibits antimicrobial properties, a study determined its potential activity against *N. fowleri*. In this study, CA significantly reduced the viability of the amoeba, both alone and conjugated with AuNp, and the conjugated form presented better amoebicidal effects against *N. fowleri* at concentrations of 50 µM as opposed to the unconjugated form, reducing the number of *N. fowleri* from 5 × 10^5^ to 1.9 × 10^5^ and 2.6 × 10^5^, respectively. CA-AuNp showed similar activity to AmB, which reduced the numbers of the amoeba to 1.6 × 10^5^. CA-AuNp showed no cytotoxicity towards Henrietta Lacks cervical cancer cells (HeLa) [86]. 

Both AuNp and AgNp have been demonstrated to increase drug bioavailability and antimicrobial activity with similar potency, as shown in a recent study conjugating Guanabenz acetate (GA) with both types of nanoparticles. GA is an FDA-approved drug capable of crossing the BBB and has demonstrated antimicrobial and anti-inflammatory activity in vitro. Although GA could reduce the viability of *N. fowleri* on its own, it required concentrations of 50 and 100 µM to produce an effect. It was reported that its conjugation with both AuNp and AgNp resulted in a significant increase in amoebicidal effects at concentrations as low as 2.5 µM. The conjugated forms also presented anti-encystment activity towards the amoeba at concentrations of 5 µM. Both conjugates presented similar amoebicidal effects and produced minimal cytotoxicity towards HeLa and HaCaT cells, but GA-AgNp presented slightly higher cytotoxicity than GA-AuNp [87]. 

Additionally, green nanoparticles synthesized with green chemistry have been studied as potentially safer and more eco-friendly drug delivery systems. Therefore, in a recent study, metal nanoparticles were stabilized with plant-derived polysaccharides, such as gum tragacanth (Gt) and gum acacia (Ga). The green nanoparticles were later conjugated with Hesperidin (HDN) and Naringin (NRG), two flavonoids that exhibit antioxidant and anti-inflammatory properties that may help reduce the immunopathogenic process unleashed during PAM. Both Ga-AgNPs-HDN and Gt-AuNPs-NRG showed significant amoebicidal activity against *N. fowleri* as opposed to the nanoparticles alone. Ga-AgNPs-HDN caused a 99% decrease in the amoeba’s viability at a concentration of 25 µg/mL, proving to be more effective than AmB. Both conjugates also reduced amoeba-mediated cell toxicity while causing no cytotoxicity to human cells [88]. Drugs conjugated with metal nanoparticles show significant potential to treat PAM and hold great promise for future in vivo evaluations.

## 8. Prevention

According to the World Health Organization (WHO), vaccination is the best and most effective way to prevent disease [89]. Therefore, a group of scientists developed a DNA vaccine that uses a lentiviral vector to produce *N. fowleri’s* Nfa1 gene. An in vivo experiment involving mice was performed to evaluate the vaccine’s effect on the amoeba. The results showed that vaccinated animals presented higher levels of IgG and higher production of IL-4 and IFN-γ, indicating a mixed-type Th1/Th2 immune response. Additionally, the vaccinated mice demonstrated a higher survival rate (90%) than the control group, in which all mice died after two weeks PI. Even though these results show that vaccination could be a preventive technique against *N. fowleri*, further studies must be done to understand its true potential to protect humans [90].

Alternatively, other preventive strategies can be carried out to avoid *N. fowleri* infection. Most PAM cases occur when people perform activities in waters infested with the amoeba, such as pools or natural bodies of water. Because *N. fowleri* is susceptible to chlorine and is killed at one part per million, the chlorination of pools and water parks is highly recommended to prevent PAM. The ameba-monitoring program developed by the South Australian High Commission routinely evaluates residual chlorine levels and total coliform counts to determine if *N. fowleri* is present in such waters. Implementing similar programs worldwide may prevent further *N. fowleri* infections. Chlorine should also be used to clean domestic water supplies, medical instruments, and hygiene devices since these have also been related to *N. fowleri* infection [25]. Furthermore, boiled water should always be used when rinsing, flushing, or irrigating nasal passages [11].

Because lakes, rivers, and ponds cannot be chlorinated, performing recreational activities such as swimming, water skiing, and diving should be avoided, especially during the summer months. Otherwise, nose plugs are encouraged to stop the amoeba from entering the nasal cavity. Digging in or stirring up the sediment found in the shallow end of these bodies of water should also be avoided. In addition, local public authorities should constantly monitor *N. fowleri’s* presence in high-risk areas. If the amoeba were to be detected, warnings should be posted to prevent people from entering the water [13].

For these preventive strategies to work, it is crucial to implement awareness campaigns in schools, colleges, religious groups, local government buildings, hospitals, and recreational places. Since there is no current treatment for PAM, taking preventative actions is the best way to avoid this deadly disease [11].

## 9. Conclusions

*N. fowleri* is one of the few FLA that is pathogenic towards humans, causing a necrotizing and hemorrhaging meningoencephalitis called PAM. Even though the amoeba causes severe tissue damage through contact-dependent and contact-independent mechanisms, it also induces a robust immune response that further injures the host. PAM is characterized by an extremely high mortality rate and causes death less than two weeks after the initial exposure. It has been suggested that an early diagnosis is crucial for a patient’s survival; however, because of lack of awareness and the clinical resemblance the infection has with viral or bacterial meningitis, most cases have been diagnosed post-mortem. Additionally, developing a safe and effective treatment has also been challenging, mainly because the infection is so rare and progresses rapidly. The current treatment regime for PAM involves AmB, combined with other drugs, but it is seldom successful and causes adverse effects. There have been different in vitro and in vivo studies determined to find a safer and more effective drug to treat PAM, some of which have shown potent amoebicidal activity. In fact, anti-inflammatory molecules may help reduce the immunopathological response that has been linked with severe tissue damage. Furthermore, the conjugation of these drugs with nanoparticles has been demonstrated to increase bioavailability, amoebicidal activity, and reduce cell toxicity in PAM infections. However, further studies are needed to confirm the true potential of these drugs. Until a more effective treatment is developed, prevention strategies are the best way to avoid PAM.

## Figures and Tables

**Figure 1 biomolecules-11-01320-f001:**
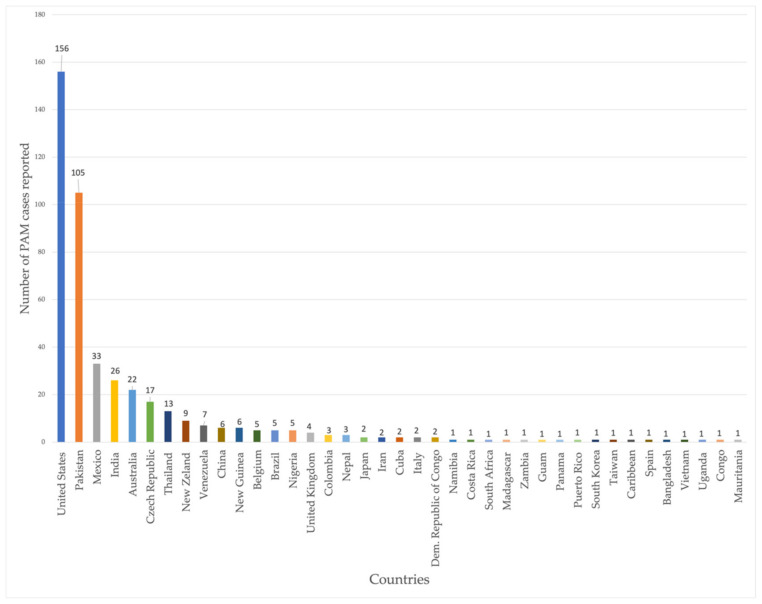
Worldwide documented cases of *N. fowleri* infections until 2018. Adapted from [11,19].

**Figure 2 biomolecules-11-01320-f002:**
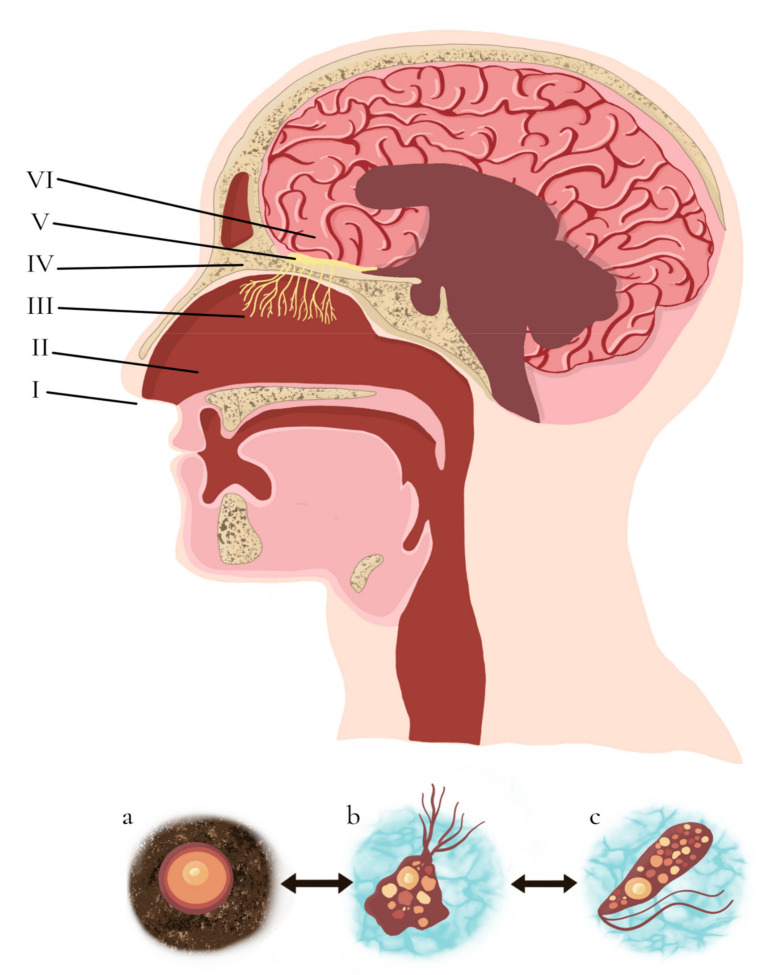
Infection mechanism and forms of *N. fowleri,* Adapted from [30]. Depending on its surrounding environment, *N. fowleri* exists in three different forms: A dormant cyst (**a**) capable of surviving different physical and chemical conditions, a reproductively active trophozoite (**b**), that can reproduce, feed, encyst, and cause infection in other organisms and a transitory flagellate (**c**) [10,11,12,13,14,15]. *N. fowleri* trophozoites infect their hosts through water entering the nasal cavity (I). Afterwards, the amoeba attaches itself to the nasal mucosa (II) and migrates along the olfactory nerves (III) through the cribriform plate (IV) until it reaches the olfactory bulb (V). Trophozoites then enter the brain (VI) through olfactory nerve bundles, where they multiply and cause severe cerebral damage and inflammation [23,28,29].

**Table 1 biomolecules-11-01320-t001:** Drugs being studied in vitro to determine their amoebicidal activity against *N. fowleri*.

Drug	Description	Proposed Mechanism of Action against *N. fowleri*	Results
Posaconazole	Anti-fungal agent	Inhibits CYP51 and blocks ergosterol production	The anti-mycotic drugs Itraconazole, Posaconazole, Ketoconazole, Isavuconazole, Miconazole, Voriconazole, Clotrimazole, and Fluconazole demonstrated anti-amoebic activity in vitro against *N. fowleri*. These drugs exhibited EC_50_ ranging from 13.9 µM (Fluconazole) to ≤0.01 µM (Itraconazole and Posaconazole). All drugs demonstrated more potency than MLT (EC_50_ 54.5 µM) but only Itraconazole and Posaconazol showed more potency than AmB (EC_50_ 0.1 µM) [68].
Isavuconazole	Anti-fungal agent	Inhibits CYP51 and blocks ergosterol production	In this study Isavuconazole and Epiminolanosterol were combined at concentrations of 0.08 µM and 0.7 µM, respectively; Isavuconazole and Tamoxifen were combined at 1.9 µM equimolar concentrations; and Epiminolanosterol and Tamoxifen were combined at 2 µM equimolar concentrations. These combinations showed synergistic activity against *N. fowleri* and managed to inhibit the amoeba’s growth by 95% [69].
Epiminolanosterol	SMT inhibitor	Inhibits SMT and blocks ergosterol production	
Tamoxifen	Antiestrogenic	Inhibits ERG2 and blocks ergosterol production	
Ebselen	Synthetic organoselenium drug	Unknown; Inhibits cysteine proteases and crosses BBB	Ebselen inhibited the growth of *N. fowleri* trophozoites by 100% at concentrations of 12.5 µM while BAY 11-7082 and BAY 11-7085 showed 100% growth inhibition with 3.12 µM. Additionally, it was reported that the EC_50_ of Ebselen, BAY 11-7082, BAY 11-7085, and MLT obtained by ATP bioluminescence method were approximately 6.2, 1.6, 2.3, and 54.5 µM, respectively. The compounds also demonstrated more potency than MLT by 8.5-fold, 34-fold, and 23-fold, respectively [70].
BAY 11-7082	Phenyl vinyl sulfone-related compound	Unknown; Inhibits cysteine proteases and crosses BBB	
BAY 11-7085	Phenyl vinyl sulfone-related compound	Unknown; Inhibits cysteine proteases and crosses BBB	
Auranofin	Antirheumatic agent	Unknown; exhibits anti-inflammatory properties and crosses the BBB	AF proved to be equally potent against five different *N. fowleri* strains (European KUL; Australian CDC:V1005; US Davis; US CAMP; US TY), with EC_50_ values ranging between 1–2 µM. After 24 h, AF inhibited the growth of the trophozoites by 97% with a concentration of 12.5 µM. Additionally, the drug showed no cytotoxicity to human astrocytes at concentrations lower than 5 µM. AF showed to be less potent than AmB but more than MLT by 15- to 60- fold [22].
Staurosporine	Indolocarbazole	Inhibiting the protein kinase and inducing an apoptosis-like mechanism	STS showed IC_50_ values of 0.08 µM for two different *N. fowleri* strains (*N. fowleri* ATCC^®^ 30808^TM^; *N. fowleri* ATCC^®^ 30215^TM^), proving to be more active than AmB and MLT. Additionally, STS demonstrated less cytotoxicity than MLT towards murine macrophage J774A.1 [71].
Etomoxir	Fatty acid oxidation inhibitors	Unknown	ETO, PHX, and TZD exhibited amoebicidal activity against *N. fowleri* trophozoites, with IC_50_ values of 146, 7.5, and 6.6 µM, respectively. On the other hand, VPA showed some inhibition against the amoeba but only at high concentrations. It was reported that the combination of MLT and PHX, MLT and TZD, MLT and VPA, PHX and TZD, TZD and VPA showed additivity, while the combination of MLT and ETO showed synergy. The synergistic activity was most noticeable when 12.5 and 25 µM concentrations of MLT were combined with 25 to 200 µM concentrations of ETO [72].
Perhexiline	Fatty acid oxidation inhibitors	Unknown	
Valproic acid	Antiepileptic drug	Unknown	
Thioridazine	Antipsychotic drug	Unknown	
Fluvastatin	Statin	Acting on sterol pathway; programed cell death	Fluvastatin and Atorvastatin exhibited amoebicidal activity against two strains of *N. fowleri* (ATCC^®^ 30808^TM^;ATCC^®^ 30215^TM^), showing IC_50_ values between 0.17–1.68 and 7.63–6.27 µM, respectively. Both statins showed better activity and less cytotoxicity than MLT but less activity and similar cytotoxicity to AmB [24,73].
Atorvastatin	Statin	Acting on sterol pathway; programed cell death	
Pitavastatin	Statin	Inhibits HMG-CoA reductase and blocks ergosterol production	Pitavastatin demonstrated anti-amoebic activity against five different *N. fowleri* strains. The drug exhibited mean EC_50_ values of 0.3 µM, 3.9 µM, 2.5 µM, 1 µM, and 4 µM for the *N. fowleri* strains European KUL, Australian CDC:V1005, US Davis (Genotype I), US CAMP (Genotype II), and US TY (Genotype III), respectively. It showed more potency than MLT but less than AmB. Finally, the study remarks that Pitavastatin is BBB permeable [74]. Additionally, a different study demonstrated Pitavastatin’s anti-amoebic activity against the *N. fowleri* strain ATCC^®^ 30808, exhibiting an IC_50_ value of 0.038 µM. It showed an IC_50_ value of 0.189 µM against the ATCC^®^ 30215 strain [75].
Cerivastatin	Statin	Acting on sterol pathway; programed cell death	Cerivastatin demonstrated an IC_50_ value of 0.113 µM and 0.064 µM for the *N. fowleri* strains ATCC^®^ 30808 and ATCC^®^ 30215 respectively. On the other hand, Rosuvastatin showed IC_50_ values of 0.264 µM and 0.821 µM for the strains ATCC^®^ 30808 and ATCC^®^ 30215 respectively [75].
Rosuvastatin	Statin	Acting on sterol pathway; programed cell death	
Lonafarnib	Farnesyltransferase inhibitor	Inhibits Farnesyltransferase and blocks ergosterol production	When evaluated against five different *N. fowleri* strains, Lonafarnib presented EC_50_ values of 1.5 µM, 2.5 µM, 5 µM, 3.5 µM and 9.2 µM for European KUL, Australian CDC:V1005, US Davis (Genotype I), US CAMP (Genotype II), and US TY (Genotype III) respectively. However, its combination with Pitavastatin showed synergistic activity, resulting in a 95% growth inhibition of the trophozoites [76].

EC_50_, Half maximum effect concentration; IC_50_, Half maximum inhibitory concentration; MLT, Miltefosine; AF, Auranofin; AmB, Amphotericin B; STS, Staurosporine; ETO, Etomoxir; PHX, Perhexiline; VPA, Valproic acid; TZD, Thioridazine.

**Table 2 biomolecules-11-01320-t002:** Drugs being studied in vivo to determine their amoebicidal activity against *N. fowleri*.

Drug	Description	Proposed Mechanism of Action against *N. fowleri*	Results
Chlorpromazine	Antipsychotic drugs	Unknown	The amoebicidal activity of CPZ against *N. fowleri* was compared to that of AmB and MLT. In vitro, AmB and CPZ managed to inhibit the amoeba’s growth by 100% at concentrations of 0.78 and 12.5 µg/mL respectively, while MLT sustained growth inhibition for six days at 25 µg/mL. Even though AmB exhibited the best activity in vitro, CPZ was the most effective in vivo, with a 75% survival rate in mice. Contrastingly, AmB and MLT presented a survival rate of 40% and 55%, respectively. Additionally, doses of 10 or 20 mg/kg of CPZ caused no liver or renal toxicity in mice [77].
Hygromycin B	Aminoglycoside antibiotic	Unknown	Seven antibiotics were evaluated to test their activity against *N. fowleri,* both in vitro and in vivo. Initially, Hy and RKM were the most effective of the seven drugs, causing a 100% inhibition at concentrations of 12.5 and 6.25 µg/mL, respectively. At the same time, ROX maintained growth inhibition at 25 µg/mL. However, Clarithromycin, Erythromycin, Neomycin, and Zeocin did not exhibit any amoebicidal activity. During in vivo evaluations, treatment with RKM showed a survival rate of 80% in infected mice, while ROX presented a survival rate of 25%. However, all mice treated with Hy died. Additionally, the study reported that a dose of 20 mg/kg of RKM induced no liver and renal toxicity in mice [78].
Rokitamycin	Semisynthetic macrolide antibiotic	Unknown	
Roxithromycin	Semisynthetic macrolide antibiotic	Unknown	
Corifungin	Polyene macrolide	Induces an apoptosis-like programmed cell death	*N. fowleri* trophozoites were treated with different concentrations of Corifungin (3.12, 6.25, 12.5, 18.75, and 25 µM) for 72 h, inhibiting the growth of the amoeba by 20, 60, 90, 95, and 100%, respectively. During in vivo evaluations, the administration of Corifungin at 9 mg/kg/day for ten days resulted in a survival rate of 100% of the infected mice and was well tolerated by the animals. Contrastingly, the same dose of AmB showed a survival rate of 60%, a value similar to the PBS controls [64].
Posaconazole	Antifungal	Unknown	After performing a high-throughput phenotypic screening method that identified various antibiotics and antifungals, it was reported that PCZ stood out the most, showing to be potent and rapidly acting. This compound was capable of inhibiting the amoeba’s growth in just 12 h and showed an IC_50_ value of 0.24 µM. PCZ also demonstrated additive activity with AZM, AmB, and MLT. Additionally, the intravenous administration of 20 mg/kg of PCZ showed a survival rate of 33% in infected mice. However, the combination of PCZ with AZM was the most effective and prolonged the survival of mice significantly [79].

CPZ, Chlorpromazine; Hy, Hygromycin B; RKM, Rokitamycin; ROX, Roxithromycin; PCZ, Posaconazole; PBS, Phosphate-buffered saline; AZM, Azithromycin.

**Table 3 biomolecules-11-01320-t003:** Drugs conjugated with metal nanoparticles being studied to determine their amoebicidal activity against *N. fowleri*.

Drug	Description	Conjugation	Proposed Mechanism of Action against *N. fowleri*	Results
Amphotericin B	Antifungal	AgNp	Induces an apoptosis-like programmed cell death	While unconjugated AmB and NYS reduced the number of *N. fowleri* trophozoites from 9.2 × 10^5^ to 3.7 × 10^5^ and 7.5 × 10^5,^ respectively, their conjugation to AgNp further increased their amoebicidal activity, reducing the number of amoebas to 2 × 10^5^ for AmB-AgNp and 5.8 × 10^5^ for NYS-AgNp [63].
Nystatin	Antifungal	AgNp	Unknown	
Aryl quinazolinone	Quinazolinone	AgNp	Unknown	Out of the 34 synthesized aryl quinazoline derivatives, Q1, Q2, Q4, Q5, Q7, Q8, Q10–Q12, Q14−Q21, Q23, Q24, and Q27−Q33 reduced the viability of *N. fowleri* trophozoites. Q23 exhibited the highest amoebicidal activity, reducing viable cells by 85%. On the other hand, Q14, Q20, Q29, and Q33 were the compounds that inhibited the amoeba’s growth. Additionally, the activity of Q12, Q13, Q15, Q18, Q24, and Q34 was evaluated before and after conjugation with AgNp; however, only Q24′s amoebicidal activity was enhanced after the conjugation, increasing it from 37 to 53%. Most of the quinazolinones exhibited limited cytotoxicity (<25%) towards HaCaT cells [81].
Oleic acid	Fatty acid	AgNp	Apoptosis	The concentration of 5 µM of OA-AgNp reduced the number of *N. fowleri* from 6.1 × 10^5^ to 2.6 × 10^5^. However, at concentrations of 10 µM, the viability of the amoeba was further reduced to 2.1 × 10^5^. This conjugation exhibited higher amoebicidal activity than AmB. Furthermore, the concentration of 10 µM of OA-AgNp decreased the amoebas cytotoxicity towards HeLa cells by 70%. Additionally, OA-AgNp produced low levels of cytotoxicity (<20%) towards HeLa cells [80].
Diazepam	Benzodiazepine	AgNp	Alteration to mitochondrial activity	5 × 10^5^ *N. fowleri* trophozoites were incubated with DZP, PBT and PTN both on their own and conjugated with AgNp. Those conjugated with AgNp exhibited enhanced amoebicidal activity compared to the drugs alone. Only PTN, alone and conjugated presented an activity equal to AmB [82].
Phenobarbitone	Barbiturate	AgNp	Inhibits the sodium/calcium channel	
Phenytoin	Anti-seizure drug	AgNp	Inhibits the sodium/calcium channel	
A1 and A2	Benzimidazole derivatives	AgNp	Unknown, probably acting on sterol pathway	Azoles A1-A6 demonstrated both anti-amoebic and amoebistatic activities when used in concentrations of 50 µM. The compounds A4 and A5 exhibited the highest amoebicidal activity, resulting in 72% and 66% cell death, respectively, while A6, A3, and A2 resulted in a cell death percentage of 45%, 39%, and 31%, respectively. A1 presented no amoebicidal activity. Azole A3 showed the highest amoebistatic activity, causing 75% growth inhibition, while A6, A2, A5, A4, and A1 resulted in 67%, 64%, 63%, 55%, and 20% amoebistatic activity. In addition, most of the azoles showed none or little cytotoxicity towards HaCaT cells. Conjugation with AgNp slightly improved the amoebicidal activity of the azoles, as it only increased A1 by 36%, and A6 by 6% [83].
A3 and A4	Indazole derivatives	AgNp	Unknown, probably acting on sterol pathway	
A5 and A6	Tetrazole derivatives	AgNp	Unknown, probably acting on sterol pathway	
A1 and A2	Bisindole derivatives	AgNp	Unknown, probably acting on sterol pathway	Azoles A1-A4 demonstrated both amoebicidal and amoebistatic activities when used in concentrations of 50 µM. The compounds A4 and A2 exhibited the highest amoebicidal activity, resulting in 69% and 53% cell death, respectively. However, no amoebicidal effect was observed for A1 and A3. On the other hand, A4, A3, and A1 resulted in 68%, 61%, and 49% growth inhibition, while no amoebistatic effect was observed for A2. In addition, the azoles A2-A4 showed none or little cytotoxicity towards HeLa cells; however, A1 proved to be highly cytotoxic (98%). Finally, conjugation with AgNp improved the amoebicidal activity of the azoles A1 by 32% and A3 by 51% [84].
A3 and A4	Thiazole derivatives	AgNp	Unknown, probably acting on sterol pathway	
Curcumin	Polyphenol	AuNp	Unknown; anti-inflammatory properties and inhibits lipid peroxidation	Curcumin demonstrated a concentration-dependent activity, where 6.25, 12.5, 25 and 200 μM of curcumin resulted in 22, 30, 35 and 66% amoebicidal activity. Its conjugation with AuNp increased its bioavailability, resulting in a 69% amoebicidal activity at a concentration of 10 µM. Additionally, concentrations of 5 and 10 µM did not exhibit any cytotoxic activities against HaCaT cells [85].
Trans-cinnamic acid	Organic acid	AuNp	Inhibits the expression of cell proliferation genes and/or blocks the post-translational modification of cell-growth-regulating proteins	*N. fowleri* was treated with different concentrations (2.5–50 µM) of CA, both alone and conjugated with AuNp. CA demonstrated a concentration-dependent amoebicidal activity. At a concentration of 50 µM, CA-AuNp exhibited better effects against the amoeba, reducing its number from 5 × 10^5^ to 1.9 × 10^5^ while the unconjugated form reduced the amoeba to 2.6 × 10^5^. Additionally, CA-AuNp presented effects similar to AmB and exhibited no cytotoxicity towards HeLa cells [86].
Guanabenz acetate	Antihypertensive drug	AgNp and AuNp	Unknown; anti-inflammatory properties and crosses BBB	GA conjugated with both AgNp and AuNp demonstrated powerful amoebicidal and anti-encystment effects against *N. fowleri* at concentrations as low as 2.5 µM and 5 µM, respectively. Contrastingly, GA on its own required concentrations from 50 to 100 µM to produce any effect. AuNp conjugates exhibited no cytotoxicity towards HeLa and HaCaT cells, although silver nanoconjugates presented modest levels of toxicity (<20%) on both cell lines [87].
Hesperidin	Flavonoid	AgNp and Ga	Unknown; presents anti-inflammatory and antioxidant properties	Both conjugates exhibited significant amoebicidal activity against *N. fowleri* as opposed to the nanoparticles alone. The HDN conjugate caused a 99% decrease in the amoeba’s viability at a concentration of 25 µg/mL, proving to be more effective than AmB. At concentrations higher than 100 µg/mL, HDN and NRG conjugates exhibited 11% and 23% cytotoxicity towards HeLa cells, respectively [88].
Naringin	Flavonoid	AuNp and Gt	Unknown; presents anti-inflammatory and antioxidant potential	

AgNp, Silver nanoparticles; AuNp, Gold nanoparticles; NYS, Nystatin; Q, Aryl quinazolinone; HaCaT, Human keratinized skin cells; OA, Oleic acid; HeLa, Henrietta Lacks cervical cancer cells; DZP, Diazepam; PBT, Phenobarbitone; PTN, Phenytoin; CA, trans-cinnamic acid; GA, Guanabenz acetate; HDN, Hesperidin; Ga, Gum acacia; NRG, Naringin; Gt, Gum tragacanth.

## Data Availability

No datasets were generated or analyzed during the current study.

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
