# Peer review of "Primary Amoebic Meningoencephalitis by Naegleria fowleri: Pathogenesis and Treatments"

_biomolecules, 2021, doi:10.3390/biom11091320_

Round 1

Reviewer 1 Report

Dear Authors, 

Naegleria fowleri is pathogenic amoeba causing an acute and fulminating infection known as primary amoebic meningoencephalitis (PAM). Currently, it is difficult to diagnose an NF infection, which most often results in death. This work is interesting and useful for the scientific community working on NF and PAM. However, this study has some weaknesses that need to be improved to be published. I have detailed below all my comments and modifications to be made in order to improve the quality of the manuscript.

Best regards

Major

Are NFs only at the origin of PAM? They are also at the origin of skin infections or others…. Specify at least in the introduction.

Please specify in the introduction or in the first paragraph the phylogeny of NF, in particular the class, genus, ect….

Specify it in the manuscript: Which species of Naegleria have already been described as pathogenic for humans, which genus and species amoebas are close to naegleria (and NF).

Certain regions of the world (India, Pakistan, USA ..) are more affected than others by NF infections, specify and develop

Is it possible to isolate nf from human samples, how they are grown in axenic and non-axenic media, specify in your manuscript (paragraph of the diagnosis)

How could we improve the diagnosis (carry out the culture of each sample ??) please develop.

Develop a small section on the prevention that can be done to avoid infections with NF (cf: Naegleria fowleri: Sources of infection, pathophysiology, diagnosis, and management; a review).

Minor:

Line 22

Space forget

« They havevarious »

Line 27 : beware some fla are partially aerobic; ex: A. castellanii (ex: Evidence for a Hydrogenosomal-Type Anaerobic ATP Generation Pathway in Acanthamoeba castellanii)

“FLA are aerobic, mitochondriate and eukaryotic microorganisms that can complete their life cycle as 28 parasites or inhabiting natural environments as free-living amoeba”

Line 29:

Repetition: FOR

For For this reason

Reviewer 2 Report

Naegleria fowleri is cosmopolitan free-living amoeba that can invade the central nervous system. Every year, there are a few case reports concerning N. fowleri infection in people who were swimming or diving in the warm outdoors water. Insufficicent knowledge about the pathogenicity, diagnosis and treatmenf of N. fowleri infections causes that the infection is characterized by high mortality rate. Therefore, I believe that the paper “Primary amoebic meningoencephalitis by Naegleria fowleri: pathogenesis and treatments” is very important.

However, some points need to be clarified and changed:
Line 8: delete “(N. fowleri)”, please
Line 22: change “havevarious” into “have various”
line 29: double "for"
Line 31: delete “living”
Line 32: “humane objects” what authors had in mind?
Line 34: delete “(N. fowleri)”
Line 35: change “produces” into “causes”
Sentence from line 37 to 39 needs to be changed (English correction)
Line 40: after sentence “studies have suggestes that an early….” you should put citation
Line 47: delete “living”
Line 50: what authors have in mind by writing “contamined water”? tap water?
Line 78: delete “the amoeba”
Line 102: people are not only diagnosed with PAM post- mortem!! They are mostly diagnosed post-mortem. In scientific literature, there are some cases of survivors.
Line 123:  please add “presence/ infection” after “further confirmation of the amoeba”
Line 130: change “largenumber” into “large number”
Line 133: “while glucose may present values from 10 mg/100 ml” to…?
Line 137: change “patiet” into “patient”
4. Diagnosis: what about the culture method?
Line 143; it is important to write “therefore, the amoeba usually infects its host…”? The sentence above has the same information.
Line 184: “componets” or components?
Line 187: please rewrite the sentence “An in vivoexperiments demonstrated that amoebas that passed… “
Line 190: please change “particularprotein” to “ particular protein”
Line 202: delete “Naegleria gruberi”. Leave only abbreviation N. gruberi
In contact-independent mechanism: please describe also the effect of MMPs, which N. fowleri has in the trophozoite form. It is described in: Kot, K.; Łanocha-Arendarczyk, N.; Kosik-Bogacka, D. Immunopathogenicity of Acanthamoeba spp. in the Brain and Lungs. Int. J. Mol. Sci. 202122, 1261. The orginal paper: Lam, C.; Jamerson, M.; Cabral, G.; Carlesso, A.M.; Marciano-Cabral, F. Expression of matrix metalloproteinases in Naegleria fowleri and their role in invasion of the central nervous system. Microbiology 2017163, 1436–1444. 

Line 437: please, rewrite the sentence ”One study demostarted the effect Chlorpromazine….”

Reviewer 3 Report

The ms presents a review on Naegleria fowleri and its therapeutics. Even though some parts have been discussed in recent papers from other authors, the therapeutics section is of interest. However, I consider that this part should be extended and data should be updated. Please revise and resubmit-

Round 2

Reviewer 1 Report

Dear Authors, 

all requested modifications have been made, the manuscript can be accepted

Best regards